# DeepSeek-R1 incentivizes reasoning in LLMs through reinforcement learning

General reasoning represents a long-standing and formidable challenge in artificial intelligence (AI). Recent breakthroughs, exemplified by large language models (LLMs)[1,2] and chain-of-thought (CoT) prompting[3], have achieved considerable success on foundational reasoning tasks. However, this success is heavily contingent on extensive human-annotated demonstrations and the capabilities of models are still insufficient for more complex problems. Here we show that the reasoning abilities of LLMs can be incentivized through pure reinforcement learning (RL), obviating the need for human-labelled reasoning trajectories. The proposed RL framework facilitates the emergent development of advanced reasoning patterns, such as self-reflection, verification and dynamic strategy adaptation. Consequently, the trained model achieves superior performance on verifiable tasks such as mathematics, coding competitions and STEM fields, surpassing its counterparts trained through conventional supervised learning on human demonstrations. Moreover, the emergent reasoning patterns exhibited by these large-scale models can be systematically used to guide and enhance the reasoning capabilities of smaller models.

Reasoning capability, the cornerstone of human intelligence, enables complex cognitive tasks ranging from mathematical problem-solving to logical deduction and programming. Recent advances in AI have demonstrated that LLMs can exhibit emergent behaviours, including reasoning abilities, when scaled to a sufficient size[4,5]. However, achieving such capabilities in pre-training typically demands substantial computational resources. In parallel, a complementary line of research has demonstrated that LLMs can be effectively augmented through CoT prompting. This technique, which involves either providing carefully designed few-shot examples or using minimalistic prompts such as "Let's think step by step"[3,6], enables models to produce intermediate reasoning steps, thereby substantially enhancing their performance on complex tasks. Similarly, further performance gains have been observed when models learn high-quality, multistep reasoning trajectories during the post-training phase[2,7]. Despite their effectiveness, these approaches exhibit notable limitations. Their dependence on human-annotated reasoning traces slows scalability and introduces cognitive biases. Furthermore, by constraining models to replicate human thought processes, their performance is inherently capped by the human-provided exemplars, which prevents the exploration of superior, non-human-like reasoning pathways.

To tackle these issues, we aim to explore the potential of LLMs for developing reasoning abilities through self-evolution in a RL framework, with minimal reliance on human labelling efforts. Specifically, we build on DeepSeek-V3 Base[8] and use Group Relative Policy Optimization (GRPO)[9] as our RL framework. The reward signal is only based on the correctness of final predictions against ground-truth answers, without imposing constraints on the reasoning process itself. Notably, we bypass the conventional supervised fine-tuning (SFT) phase before RL training. This design choice originates from our hypothesis that human-defined reasoning patterns may limit model exploration, whereas unrestricted RL training can better incentivize the emergence of new reasoning capabilities in LLMs. Through this process, detailed in the next section, our model (referred to as DeepSeek-R1-Zero) naturally developed diverse and sophisticated reasoning behaviours. To solve reasoning problems, the model exhibits a tendency to generate longer responses, incorporating verification, reflection and the exploration of alternative approaches within each response. Although we do not explicitly teach the model how to reason, it successfully learns improved reasoning strategies through RL.

Although DeepSeek-R1-Zero demonstrates excellent reasoning capabilities, it faces challenges such as poor readability and language mixing, occasionally combining English and Chinese in a single CoT response. Furthermore, the rule-based RL training stage of DeepSeek-R1-Zero is narrowly focused on reasoning tasks, resulting in limited performance in broader areas such as writing and open-domain question answering. To address these challenges, we introduce DeepSeek-R1, a model trained through a multistage learning framework that integrates rejection sampling, RL and supervised fine-tuning, detailed in the 'DeepSeek-R1' section. This training pipeline enables DeepSeek-R1 to inherit the reasoning capabilities of its predecessor, DeepSeek-R1-Zero, while aligning model behaviour with human preferences through further non-reasoning data.

To enable broader access to powerful AI at a lower energy cost, we have distilled several smaller models and made them publicly available. These distilled models exhibit strong reasoning capabilities, surpassing the performance of their original instruction-tuned counterparts. We believe that these instruction-tuned versions will also greatly contribute to the research community by providing a valuable resource for understanding the mechanisms underlying long CoT reasoning models and for promoting the development of more powerful reasoning models. We release DeepSeek-R1-Zero, DeepSeek-R1, data samples and distilled models to the public as described in the 'Code availability' section.

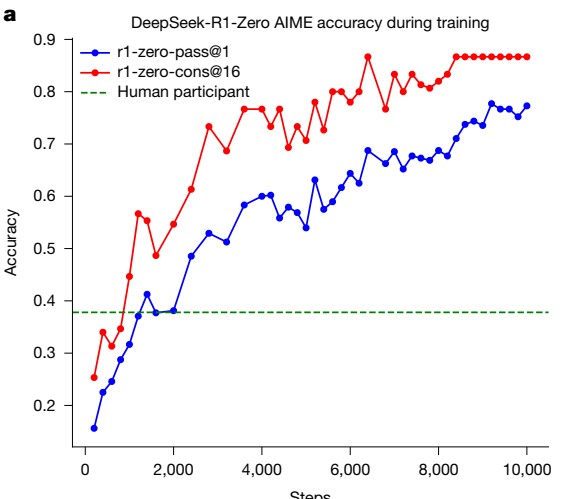

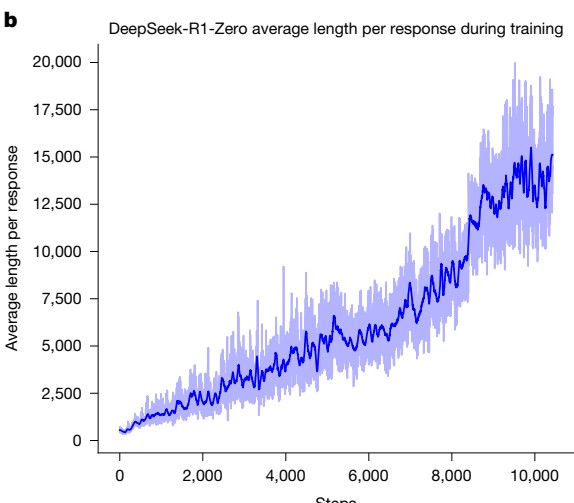

**Fig. 1 | Accuracy and output length of DeepSeek-R1-Zero throughout the training process. a**, AIME accuracy of DeepSeek-R1-Zero during training. AIME takes a mathematical problem as input and a number as output, illustrated in Extended Data Table 1. pass@1 and cons@16 are described in Supplementary Information, section 4.1. The baseline is the average score achieved by human participants in the AIME competition. **b**, The average response length of DeepSeek-R1-Zero on the training set during the RL process. DeepSeek-R1-Zero naturally learns to solve reasoning tasks with more thinking time. Note that a training step refers to a single policy update operation.

## DeepSeek-R1-Zero

To implement large-scale RL of DeepSeek-R1-Zero, we use a highly efficient RL pipeline. Specifically, we use GRPO[9] as our RL algorithm, described in Methods section 'GRPO'. Furthermore, we use a rule-based reward system to compute accuracy and format rewards, with detailed methodologies outlined in Methods section 'Reward design'. Furthermore, our high-performance RL infrastructure is described in Supplementary Information, section 2.1, ensuring scalable and efficient training.

Specifically, we apply the RL technique on the DeepSeek-V3 Base to train DeepSeek-R1-Zero. During training, we design a straightforward template to require DeepSeek-R1-Zero to first produce a reasoning process, followed by the final answer. The prompt template is written as below.

"A conversation between User and Assistant. The User asks a question and the Assistant solves it. The Assistant first thinks about the reasoning process in the mind and then provides the User with the answer. The reasoning process and answer are enclosed within <think>...</think> and <answer>...</answer> tags, respectively, that is, <think> reasoning process here </think><answer> answer here </answer>. User: prompt. Assistant:", in which the prompt is replaced with the specific reasoning question during training. We intentionally limit our constraints to this structural format, avoiding any content-specific biases to ensure that we can accurately observe the natural progression of the model during the RL process.

Figure 1a shows the performance trajectory of DeepSeek-R1-Zero on the American Invitational Mathematics Examination (AIME) 2024 benchmark throughout the RL training process, in which the average pass@1 score on AIME 2024 shows a marked increase, jumping from an initial value of 15.6% to 77.9%. Also, by using the self-consistency decoding[10], the performance of the model can be further improved, achieving an accuracy of 86.7%. This performance greatly surpasses the average performance across all human competitors of the AIME. Besides the maths competitions, as shown in Supplementary Fig. 8, DeepSeek-R1-Zero also achieves remarkable performance in coding competitions and graduate-level biology, physics and chemistry problems. These results underscore the effectiveness of RL in enhancing the reasoning capabilities of LLMs.

As well as the progressive enhancement of reasoning capabilities during training, DeepSeek-R1-Zero also demonstrates self-evolutionary behaviour with RL training. As shown in Fig. 1b, DeepSeek-R1-Zero exhibits a steady increase in thinking time throughout training, driven only by intrinsic adaptation rather than external modifications. Making use of long CoT, the model progressively refines its reasoning, generating hundreds to thousands of tokens to explore and improve its problem-solving strategies.

The increase in thinking time helps with the autonomous development of sophisticated behaviours. Specifically, DeepSeek-R1-Zero increasingly exhibits advanced reasoning strategies such as reflective reasoning and systematic exploration of alternative solutions provided in Extended Data Fig. 1a, substantially boosting its performance on verifiable tasks such as maths and coding. Notably, during training, DeepSeek-R1-Zero exhibits an 'aha moment', shown in Table 1, characterized by a sudden increase in the use of the word 'wait' during reflections, provided in Extended Data Fig. 1b. This moment marks a distinct change in reasoning patterns and clearly shows the self-evolution process of DeepSeek-R1-Zero.

The self-evolution of DeepSeek-R1-Zero underscores the power and beauty of RL: rather than explicitly teaching the model how to solve a problem, we simply provide it with the right incentives and it autonomously develops advanced problem-solving strategies. This serves as a reminder of the potential of RL to unlock higher levels of capabilities in LLMs, paving the way for more autonomous and adaptive models in the future.

## DeepSeek-R1

Although DeepSeek-R1-Zero exhibits strong reasoning capabilities, it faces several issues. DeepSeek-R1-Zero struggles with challenges such as poor readability and language mixing, as DeepSeek-V3 Base is trained on several languages, especially English and Chinese. To address these issues, we develop DeepSeek-R1, whose pipeline is illustrated in Fig. 2. In the initial stage, we collect thousands of cold-start data that exhibit a conversational, human-aligned thinking process, as detailed in Supplementary Information, section 2.3.2. RL training is then applied with hyperparameters in Methods section 'Training details of the first RL stage', data details in Supplementary Information, section 2.3.1, to improve the model performance with the

conversational thinking process and language consistency. Subsequently, we apply rejection sampling and SFT once more. This stage incorporates both reasoning and non-reasoning datasets into the SFT process, as detailed in Supplementary Information, section 2.3.3, enabling the model to not only excel in reasoning tasks but also demonstrate advanced writing capabilities. To further align the model with human preferences, we implement a secondary RL stage designed to enhance the helpfulness and harmlessness of the model while simultaneously refining its reasoning capabilities. The reward model is described in Methods section 'Reward design' and RL hyperparameters are in Methods section 'Training details of the second RL stage'. The total training cost is listed in Supplementary Information, section 2.4.4.

We evaluate our models on MMLU[11], MMLU-Redux[12], MMLU-Pro[13], DROP[14], C-Eval[15], IFEval[16], FRAMES[17], GPQA Diamond[18], SimpleQA[19], C-SimpleQA[20], CLUEWSC[21], AlpacaEval 2.0 (ref. 22), Arena-Hard[23], SWE-bench Verified[24], Aider-Polyglot[25], LiveCodeBench[26] (2024-08–2025-01), Codeforces[27], Chinese National High School Mathematics Olympiad (CNMO 2024)[28] and AIME 2024 (ref. 29). The details of these benchmarks are provided in Supplementary Tables 15–29.

Table 2 summarizes the performance of DeepSeek-R1 across several developmental stages, as outlined in Fig. 2. A comparison between DeepSeek-R1-Zero and DeepSeek-R1 Dev1 reveals substantial improvements in instruction-following, as evidenced by higher scores on the IF-Eval and Arena-Hard benchmarks. However, owing to the limited size of the cold-start dataset, Dev1 exhibits a partial degradation in reasoning performance compared with DeepSeek-R1-Zero, most notably on the AIME benchmark. By contrast, DeepSeek-R1 Dev2 demonstrates marked performance enhancements on benchmarks that require advanced reasoning skills, including those focused on code generation, mathematical problem solving and STEM-related tasks. Benchmarks targeting general-purpose tasks, such as Alpac-aEval 2.0, show marginal improvement. These results indicate that reasoning-oriented RL considerably enhances reasoning capabilities while exerting limited influence on user-preference-oriented benchmarks.

DeepSeek-R1 Dev3 integrates both reasoning and non-reasoning datasets into the SFT pipeline, thereby enhancing the proficiency of the model in both reasoning and general language-generation tasks. Compared with Dev2, DeepSeek-R1 Dev3 achieves notable performance improvements on AlpacaEval 2.0 and Aider-Polyglot, attributable to the inclusion of large-scale non-reasoning corpora and code-engineering datasets. Finally, comprehensive RL training on DeepSeek-R1 Dev3 using mixed reasoning-focused and general-purpose data produced the final DeepSeek-R1. Marginal improvements occurred in code and mathematics benchmarks, as substantial reasoning-specific RL was done in previous stages. The primary advancements in the final DeepSeek-R1 were in general instruction-following and user-preference benchmarks, with AlpacaEval 2.0 improving by 25% and Arena-Hard by 17%.

We also compare DeepSeek-R1 with other models in Supplementary Information, section 4.2. Model safety evaluations are provided in Supplementary Information, section 4.3. A comprehensive analysis of evaluation is provided in Supplementary Information, section 5, including a comparison with DeepSeek-V3, performance evaluations on both fresh test sets, a breakdown of mathematical capabilities by category and an investigation of test-time scaling behaviour. Supplementary Information, section 6 shows that the strong reasoning capability can be transferred to smaller models.

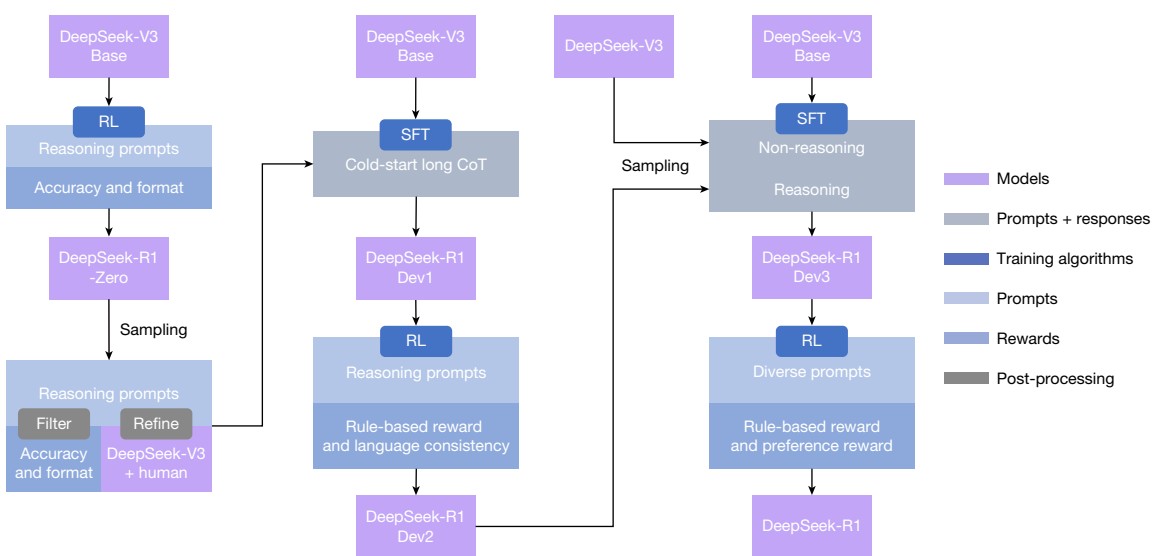

**Fig. 2 | The multistage pipeline of DeepSeek-R1.** A detailed background on DeepSeek-V3 Base and DeepSeek-V3 is provided in Supplementary Information, section 1.1. The models DeepSeek-R1 Dev1, Dev2 and Dev3 represent intermediate checkpoints in this pipeline.

**Table 2 | Experimental results at each stage of DeepSeek-R1**

| | Benchmark (metric) | R1-Zero | R1Dev1 | R1Dev2 | R1Dev3 | R1 |
|---|---|---|---|---|---|---|
| English | MMLU (EM) | 88.8 | 89.1 | **91.2** | 91.0 | 90.8 |
| | MMLU-Redux (EM) | 85.6 | 90.0 | 93.0 | 93.1 | 92.9 |
| | MMLU-Pro (EM) | 68.9 | 74.1 | 83.8 | 83.1 | **84.0** |
| | DROP (3-shot F1) | 89.1 | 89.8 | 91.1 | 88.7 | **92.2** |
| | IF-Eval (Prompt Strict) | 46.6 | 71.7 | 72.0 | 78.1 | **83.3** |
| | GPQA Diamond (Pass@1) | **75.8** | 66.1 | 70.7 | 71.2 | 71.5 |
| | SimpleQA (Correct) | 30.3 | 17.8 | 28.2 | 24.9 | 30.1 |
| | FRAMES (Acc.) | 82.3 | 78.5 | 81.8 | 81.9 | **82.5** |
| | AlpacaEval 2.0 (LC-winrate) | 24.7 | 50.1 | 55.8 | 62.1 | **87.6** |
| | Arena-Hard (GPT-4-1106) | 53.6 | 77.0 | 73.2 | 75.6 | **92.3** |
| Code | LiveCodeBench (Pass@1-COT) | 50.0 | 57.5 | 63.5 | 64.6 | **65.9** |
| | Codeforces (Percentile) | 80.4 | 84.5 | 90.5 | 92.1 | **96.3** |
| | Codeforces (Rating) | 1,444 | 1,534 | 1,687 | 1,746 | **2,029** |
| | SWE-bench Verified (Resolved) | 43.2 | 39.6 | 44.6 | 45.6 | **49.2** |
| | Aider-Polyglot (Acc.) | 12.2 | 6.7 | 25.6 | 44.8 | **53.3** |
| Maths | AIME 2024 (Pass@1) | 77.9 | 59.0 | 74.0 | 78.1 | **79.8** |
| | MATH-500 (Pass@1) | 95.9 | 94.2 | 95.9 | 95.4 | **97.3** |
| | CNMO 2024 (Pass@1) | **88.1** | 58.0 | 73.9 | 77.3 | 78.8 |
| Chinese | CLUEWSC (EM) | 93.1 | 92.8 | 92.6 | 91.6 | 92.8 |
| | C-Eval (EM) | **92.8** | 85.7 | 91.9 | 86.4 | 91.8 |
| | C-SimpleQA (Correct) | 66.4 | 58.8 | 64.2 | 66.9 | 63.7 |

Numbers in bold denote that the performance is statistically significant ($t$-test with $P<0.01$).

## Ethics and safety statement

With the advancement in the reasoning capabilities of DeepSeek-R1, we deeply recognize the potential ethical risks. For example, R1 can be subject to jailbreak attacks, leading to the generation of dangerous content such as explosive manufacturing plans, whereas the enhanced reasoning capabilities enable the model to provide plans with better operational feasibility and executability. Besides, a public model is also vulnerable to further fine-tuning that could compromise inherent safety protections.

In Supplementary Information, section 4.3, we present a comprehensive safety report from several perspectives, including performance on open-source and in-house safety evaluation benchmarks, and safety levels across several languages and against jailbreak attacks. These comprehensive safety analyses conclude that the inherent safety level of the DeepSeek-R1 model, compared with other state-of-the-art models, is generally at a moderate level (comparable with GPT-4o (2024-05-13)[30]). Besides, when coupled with the risk control system, the safety level of the model is increased to a superior standard.

## Conclusion, limitation and future work

We present DeepSeek-R1-Zero and DeepSeek-R1, which rely on large-scale RL to incentivize model reasoning behaviours. Our results demonstrate that pre-trained checkpoints inherently have substantial potential for complex reasoning tasks. We believe that the key to unlocking this potential lies not in large-scale human annotation but in the provision of hard reasoning questions, a reliable verifier and sufficient computational resources for RL. Sophisticated reasoning behaviours, such as self-verification and reflection, seemed to emerge organically during the RL process.

Even if DeepSeek-R1 achieves frontier results on reasoning benchmarks, it still faces several capability limitations, as outlined below.

## Structure output and tool use

At present, the structural output capabilities of DeepSeek-R1 remain suboptimal compared with existing models. Moreover, DeepSeek-R1 cannot make use of tools, such as search engines and calculators, to improve the performance of output. However, as it is not hard to build a RL environment for structure output and tool use, we believe that the issue will be addressed in the next version.

## Token efficiency

Unlike conventional test-time computation scaling approaches, such as majority voting or Monte Carlo tree search (MCTS), DeepSeek-R1 dynamically allocates computational resources during inference according to the complexity of the problem at hand. Specifically, it uses fewer tokens to solve simple tasks but generating more tokens for complex tasks. Nevertheless, there remains room for further optimization in terms of token efficiency, as instances of excessive reasoning—manifested as overthinking—are still observed in response to simpler questions.

## Language mixing

DeepSeek-R1 is at present optimized for Chinese and English, which may result in language-mixing issues when handling queries in other languages. For instance, DeepSeek-R1 might use English for reasoning and responses, even if the query is in a language other than English or Chinese. We aim to address this limitation in future updates. The limitation may be related to the base checkpoint, DeepSeek-V3 Base, which mainly uses Chinese and English, so that it can achieve better results with the two languages in reasoning.

## Prompting engineering

When evaluating DeepSeek-R1, we observe that it is sensitive to prompts. Few-shot prompting consistently degrades its performance. Therefore, we recommend that users directly describe the problem and specify the output format using a zero-shot setting for optimal results.

## Software-engineering tasks

Owing to the long evaluation times, which affect the efficiency of the RL process, large-scale RL has not been applied extensively in software-engineering tasks. As a result, DeepSeek-R1 has not demonstrated a huge improvement over DeepSeek-V3 on software-engineering benchmarks. Future versions will address this by implementing rejection sampling on software-engineering data or incorporating asynchronous evaluations during the RL process to improve efficiency.

Beyond specific capability limitations, the pure RL methodology itself also presents inherent challenges:

## Reward hacking

The success of pure RL depends on reliable reward signals. In this study, we ensure reward reliability through a reasoning-domain rule-based reward model. However, such dependable reward models are difficult to construct for certain tasks, such as writing. If the reward signal is assigned by a model instead of predefined rules, it becomes more susceptible to exploitation as training progresses, which means that the policy model may find shortcuts to hack the reward model. Consequently, for complex tasks that cannot be effectively evaluated by a reliable reward model, scaling up pure RL methods remains an open challenge.

In this work, for tasks that cannot obtain a reliable signal, DeepSeek-R1 uses human annotation to create supervised data and only conducts RL for hundreds of steps. We hope that, in the future, a robust reward model can be obtained to address such issues.

With the advent of pure RL methods such as DeepSeek-R1, the future holds immense potential for solving any task that can be effectively evaluated by a verifier, regardless of its complexity for humans. Machines equipped with such advanced RL techniques are poised to surpass human capabilities in these domains, driven by their ability to optimize performance iteratively through trial and error. However, challenges remain for tasks for which constructing a reliable reward model is inherently difficult. In such cases, the lack of a robust feedback mechanism may slow progress, suggesting that future research should focus on developing innovative approaches to define and refine reward structures for these complex, less verifiable problems.

Furthermore, making use of tools during the reasoning process holds notable promise. Whether it is using tools such as compilers or search engines to retrieve or compute necessary information or using external tools such as biological or chemical reagents to validate final results in the real world, this integration of tool-augmented reasoning could greatly enhance the scope and accuracy of machine-driven solutions.

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

**Daya Guo[1], Dejian Yang[1], Haowei Zhang[1], Junxiao Song[1], Peiyi Wang[1], Qihao Zhu[1], Runxin Xu[1], Ruoyu Zhang[1], Shirong Ma[1], Xiao Bi[1], Xiaokang Zhang[1], Xingkai Yu[1], Yu Wu[1], Z. F. Wu[1], Zhibin Gou[1], Zhihong Shao[1], Zhuoshu Li[1], Ziyi Gao[1], Aixin Liu[1], Bing Xue[1], Bingxuan Wang[1], Bochao Wu[1], Bei Feng[1], Chengda Lu[1], Chenggang Zhao[1], Chengqi Deng[1], Chong Ruan[1], Damai Dai[1], Deli Chen[1], Dongjie Ji[1], Erhang Li[1], Fangyun Lin[1], Fucong Dai[1], Fuli Luo[1,2], Guangbo Hao[1], Guanting Chen[1], Guowei Li[1], H. Zhang[1], Hanwei Xu[1], Honghui Ding[1], Huazuo Gao[1], Hui Qu[1], Hui Li[1], Jianzhong Guo[1], Jiashi Li[1], Jingchang Chen[1], Jingyang Yuan[1], Jinhao Tu[1,3], Junjie Qiu[1], Junlong Li[1], J. L. Cai[1], Jiaqi Ni[1], Jian Liang[1], Jin Chen[1], Kai Dong[1], Kai Hu[1,4], Kaichao You[1], Kaige Gao[1], Kang Guan[1], Kexin Huang[1,5], Kuai Yu[1], Lean Wang[1], Lecong Zhang[1], Liang Zhao[1], Litong Wang[1], Liyue Zhang[1], Lei Xu[1], Leyi Xia[1], Mingchuan Zhang[1], Minghua Zhang[1], Minghui Tang[1], Mingxu Zhou[1], Meng Li[1], Miaojun Wang[1], Mingming Li[1], Ning Tian[1], Panpan Huang[1], Peng Zhang[1], Qiancheng Wang[1], Qinyu Chen[1], Qiushi Du[1], Ruiqi Ge[1], Ruisong Zhang[1], Ruizhe Pan[1], Runji Wang[1], R. J. Chen[1],**

R. L. Jin[1], Ruyi Chen[1], Shanghao Lu[1], Shangyan Zhou[1], Shanhuang Chen[1], Shengfeng Ye[1], Shiyu Wang[1], Shuiping Yu[1], Shunfeng Zhou[1], Shuting Pan[1], S. S. Li[1], Shuang Zhou[1], Shaoqing Wu[1], Tao Yun[1], Tian Pei[1], Tianyu Sun[1], T. Wang[1], Wangding Zeng[1], Wen Liu[1], Wenfeng Liang[1✉], Wenjun Gao[1], Wenqin Yu[1,5], Wentao Zhang[1], W. L. Xiao[1], Wei An[1], Xiaodong Liu[1], Xiaohan Wang[1], Xiaokang Chen[1], Xiaotao Nie[1], Xin Cheng[1], Xin Liu[1], Xin Xie[1], Xingchao Liu[1], Xinyu Yang[1], Xinyuan Li[1,5], Xuecheng Su[1], Xuheng Lin[1], X. Q. Li[1], Xiangyue Jin[1], Xiaojin Shen[1], Xiaosha Chen[1], Xiaowen Sun[1], Xiaoxiang Wang[1], Xinnan Song[1], Xinyi Zhou[1], Xianzu Wang[1], Xinxia Shan[1], Y. K. Li[1], Y. Q. Wang[1], Y. X. Wei[1], Yang Zhang[1], Yanhong Xu[1], Yao Li[1], Yao Zhao[1], Yaofeng Sun[1], Yaohui Wang[1], Yi Yu[1], Yichao Zhang[1], Yifan Shi[1], Yiliang Xiong[1], Ying He[1], Yishi Piao[1], Yisong Wang[1], Yixuan Tan[1], Yiyang Ma[1], Yiyuan Liu[1], Yongqiang Guo[1], Yuan Ou[1], Yuduan Wang[1], Yue Gong[1,5], Yuheng Zou[1], Yujia He[1,5], Yunfan Xiong[1], Yuxiang Luo[1], Yuxiang You[1], Yuxuan Liu[1], Yuyang Zhou[1], Y. X. Zhu[1], Yanping Huang[1], Yaohui Li[1], Yi Zheng[1], Yuchen Zhu[1], Yunxian Ma[1], Ying Tang[1], Yukun Zha[1], Yuting Yan[1], Z. Z. Ren[1], Zehui Ren[1], Zhangli Sha[1], Zhe Fu[1], Zhean Xu[1], Zhenda Xie[1], Zhengyan Zhang[1], Zhewen Hao[1], Zhicheng Ma[1], Zhigang Yan[1], Zhiyu Wu[1], Zihui Gu[1], Zijia Zhu[1], Zijun Liu[1,6], Zilin Li[1], Ziwei Xie[1], Ziyang Song[1,7], Zizheng Pan[1], Zhen Huang[1], Zhipeng Xu[1], Zhongyu Zhang[1] & Zhen Zhang[1]

[1]DeepSeek-AI Team, Hangzhou, China. [2]Present address: Individual Researcher, Beijing, China. [3]Present address: Jianping High School, Shanghai, China. [4]Present address: University of Science and Technology of China, Hefei, China. [5]Present address: Peking University, Beijing, China. [6]Present address: Tsinghua University, Beijing, China. [7]Present address: Citadel Securities, Hong Kong SAR, China. ✉e-mail: wenfeng.liang@deepseek.com

## Methods

### GRPO

GRPO[9] is the RL algorithm that we use to train DeepSeek-R1-Zero and DeepSeek-R1. It was originally proposed to simplify the training process and reduce the resource consumption of proximal policy optimization (PPO)[31], which is widely used in the RL stage of LLMs[32]. The pipeline of GRPO is shown in Extended Data Fig. 2.

For each question $q$, GRPO samples a group of outputs $\{o_1, o_2,..., o_G\}$ from the old policy $\pi_{\theta_{old}}$ and then optimizes the policy model $\pi_\theta$ by maximizing the following objective:

$$\mathcal{J}_{GRPO}(\theta) = \mathbb{E}[q \sim P(Q), \{o_i\}_{i=1}^{G} \sim \pi_{\theta_{old}}(O|q)]$$

$$\frac{1}{G}\sum_{i=1}^{G}\left(\min\left(\frac{\pi_\theta(o_i|q)}{\pi_{\theta_{old}}(o_i|q)}A_i, \text{ clip}\left(\frac{\pi_\theta(o_i|q)}{\pi_{\theta_{old}}(o_i|q)}, 1-\epsilon, 1+\epsilon\right)A_i\right)\right. \tag{1}$$

$$\left. - \beta\mathbb{D}_{KL}(\pi_\theta||\pi_{ref})\right),$$

$$\mathbb{D}_{KL}(\pi_\theta||\pi_{ref}) = \frac{\pi_{ref}(o_i|q)}{\pi_\theta(o_i|q)} - \log\frac{\pi_{ref}(o_i|q)}{\pi_\theta(o_i|q)} - 1, \tag{2}$$

in which $\pi_{ref}$ is a reference policy, $\epsilon$ and $\beta$ are hyperparameters and $A_i$ is the advantage, computed using a group of rewards $\{r_1, r_2,..., r_G\}$ corresponding to the outputs in each group:

$$A_i = \frac{r_i - \text{mean}(\{r_1, r_2, \cdots, r_G\})}{\text{std}(\{r_1, r_2, \cdots, r_G\})}. \tag{3}$$

We give a comparison of GRPO and PPO in Supplementary Information, section 1.3.

### Reward design

The reward is the source of the training signal, which decides the direction of RL optimization. For DeepSeek-R1-Zero, we use rule-based rewards to deliver precise feedback for data in mathematical, coding and logical reasoning domains. For DeepSeek-R1, we extend this approach by incorporating both rule-based rewards for reasoning-oriented data and model-based rewards for general data, thereby enhancing the adaptability of the learning process across diverse domains.

**Rule-based rewards.** Our rule-based reward system mainly consists of two types of reward: accuracy rewards and format rewards.

Accuracy rewards evaluate whether the response is correct. For example, in the case of maths problems with deterministic results, the model is required to provide the final answer in a specified format (for example, within a box), enabling reliable rule-based verification of correctness. Similarly, for code competition prompts, a compiler can be used to evaluate the responses of the model against a suite of predefined test cases, thereby generating objective feedback on correctness.

Format rewards complement the accuracy reward model by enforcing specific formatting requirements. In particular, the model is incentivized to encapsulate its reasoning process within designated tags, specifically <think> and </think>. This ensures that the thought process of the model is explicitly delineated, enhancing interpretability and facilitating subsequent analysis.

$$\text{Reward}_{rule} = \text{Reward}_{acc} + \text{Reward}_{format} \tag{4}$$

The accuracy, reward and format reward are combined with the same weight. Notably, we abstain from applying neural reward models—whether outcome-based or process-based—to reasoning tasks. This decision is predicated on our observation that neural reward models are susceptible to reward hacking during large-scale RL.

Moreover, retraining such models necessitates substantial computational resources and introduces further complexity into the training pipeline, thereby complicating the overall optimization process.

**Model-based rewards.** For general data, we resort to reward models to capture human preferences in complex and nuanced scenarios. We build on the DeepSeek-V3 pipeline and use a similar distribution of preference pairs and training prompts. For helpfulness, we focus exclusively on the final summary, ensuring that the assessment emphasizes the use and relevance of the response to the user while minimizing interference with the underlying reasoning process. For harmlessness, we evaluate the entire response of the model, including both the reasoning process and the summary, to identify and mitigate any potential risks, biases or harmful content that may arise during the generation process.

**Helpful reward model.** For helpful reward model training, we first generate preference pairs by prompting DeepSeek-V3 using the Arena-Hard prompt format, listed in Supplementary Information, section 2.2, for which each pair consists of a user query along with two candidate responses. For each preference pair, we query DeepSeek-V3 four times, randomly assigning the responses as either Response A or Response B to mitigate positional bias. The final preference score is determined by averaging the four independent judgments, retaining only those pairs for which the score difference (Δ) exceeds 1 to ensure meaningful distinctions. Furthermore, to minimize length-related biases, we ensure that the chosen and rejected responses of the whole dataset have comparable lengths. In total, we curated 66,000 data pairs for training the reward model. The prompts used in this dataset are all non-reasoning questions and are sourced either from publicly available open-source datasets or from users who have explicitly consented to share their data for the purpose of model improvement. The architecture of our reward model is consistent with that of DeepSeek-R1, with the addition of a reward head designed to predict scalar preference scores.

$$\text{Reward}_{helpful} = \text{RM}_{helpful}(\text{Response}_A, \text{Response}_B) \tag{5}$$

The helpful reward models were trained with a batch size of 256, a learning rate of $6 \times 10^{-6}$ and for a single epoch over the training dataset. The maximum sequence length during training is set to 8,192 tokens, whereas no explicit limit is imposed during reward model inference.

**Safety reward model.** To assess and improve model safety, we curated a dataset of 106,000 prompts with model-generated responses annotated as 'safe' or 'unsafe' according to predefined safety guidelines. Unlike the pairwise loss used in the helpfulness reward model, the safety reward model was trained using a pointwise methodology to distinguish between safe and unsafe responses. The training hyperparameters are the same as the helpful reward model.

$$\text{Reward}_{safety} = \text{RM}_{safety}(\text{Response}) \tag{6}$$

For general queries, each instance is categorized as belonging to either the safety dataset or the helpfulness dataset. The general reward, $\text{Reward}_{general}$, assigned to each query corresponds to the respective reward defined in the associated dataset.

### Training details

**Training details of DeepSeek-R1-Zero.** To train DeepSeek-R1-Zero, we set the learning rate to $3 \times 10^{-6}$, the Kullback–Leibler (KL) coefficient to 0.001 and the sampling temperature to 1 for rollout. For each question, we sample 16 outputs with a maximum length of 32,768 tokens before the 8.2k step and 65,536 tokens afterward. As a result, both the performance and response length of DeepSeek-R1-Zero exhibit a substantial jump at the 8.2k step, with training continuing for a total of 10,400 steps, corresponding to 1.6 training epochs. Each training step consists of 32 unique questions, resulting in a training batch size of 512 per step. Every 400 steps, we replace the reference model with

the latest policy model. To accelerate training, each rollout generates 8,192 outputs, which are randomly split into 16 minibatches and trained for only a single inner epoch.

**Training details of the first RL stage.** In the first stage of RL, we set the learning rate to $3 \times 10^{-6}$, the KL coefficient to 0.001, the GRPO clip ratio $\epsilon$ to 10 and the sampling temperature to 1 for rollout. For each question, we sample 16 outputs with a maximum length of 32,768. Each training step consists of 32 unique questions, resulting in a training batch size of 512 per step. Every 400 steps, we replace the reference model with the latest policy model. To accelerate training, each rollout generates 8,192 outputs, which are randomly split into 16 minibatches and trained for only a single inner epoch. However, to mitigate the issue of language mixing, we introduce a language consistency reward during RL training, which is calculated as the proportion of target language words in the CoT.

$$\text{Reward}_{\text{language}} = \frac{\text{Num}(\text{Words}_{\text{target}})}{\text{Num}(\text{Words})} \tag{7}$$

Although ablation experiments in Supplementary Information, sction 2.6 show that such alignment results in a slight degradation in the performance of the model, this reward aligns with human preferences, making it more readable. We apply the language consistency reward to both reasoning and non-reasoning data by directly adding it to the final reward.

Note that the clip ratio plays a crucial role in training. A lower value can lead to the truncation of gradients for a large number of tokens, thereby degrading the performance of the model, whereas a higher value may cause instability during training. Details of RL data used in this stage are provided in Supplementary Information, section 2.3.

**Training details of the second RL stage.** Specifically, we train the model using a combination of reward signals and diverse prompt distributions. For reasoning data, we follow the methodology outlined in DeepSeek-R1-Zero, which uses rule-based rewards to guide learning in mathematical, coding and logical reasoning domains. During the training process, we observe that CoT often exhibits language mixing, particularly when RL prompts involve several languages. For general data, we use reward models to guide training. Ultimately, the integration of reward signals with diverse data distributions enables us to develop a model that not only excels in reasoning but also assigns priority to helpfulness and harmlessness. Given a batch of data, the reward can be formulated as

$$\text{Reward} = \text{Reward}_{\text{reasoning}} + \text{Reward}_{\text{general}} + \text{Reward}_{\text{language}} \tag{8}$$

in which

$$\text{Reward}_{\text{reasoning}} = \text{Reward}_{\text{rule}} \tag{9}$$

$$\text{Reward}_{\text{general}} = \text{Reward}_{\text{reward\_model}} + \text{Reward}_{\text{format}} \tag{10}$$

The second stage of RL retains most of the parameters from the first stage, with the key difference being a reduced temperature of 0.7, as we find that higher temperatures in this stage lead to incoherent generation. The stage comprises a total of 1,700 training steps, during which general instruction data and preference-based rewards are incorporated exclusively in the final 400 steps. We find that more training steps with the model-based preference reward signal may lead to reward hacking, which is documented in Supplementary Information, section 2.5.

## Data availability

We provide the data samples used in our rejection sampling and RL prompts at https://github.com/deepseek-ai/DeepSeek-R1 (https://doi.org/10.5281/zenodo.15753193)[33]. Comprehensive statistics and details of our complete data-generation methodology are presented in Supplementary Information, section 2.3.

## Code availability

Trained weights of DeepSeek-R1-Zero and DeepSeek-R1 are available under an MIT license at https://github.com/deepseek-ai/DeepSeek-R1 (https://doi.org/10.5281/zenodo.15753193)[33]. The inference script is released at https://github.com/deepseek-ai/DeepSeek-V3 (https://doi.org/10.5281/zenodo.15753347)[34]. Neural networks were developed with PyTorch[35] and the distributed framework is based on our internal framework HAI-LLM (https://www.high-flyer.cn/en/blog/hai-llm). The inference framework is based on vLLM[36]. Data analysis used Python v.3.8 (https://www.python.org/), NumPy v.1.23.1 (https://github.com/numpy/numpy), Matplotlib v.3.5.2 (https://github.com/matplotlib/matplotlib) and TensorBoard v.2.9.1 (https://github.com/tensorflow/tensorboard).

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

**Acknowledgements** The research is supported by DeepSeek-AI.

**Author contributions** All authors have contributed to the publication, being variously involved in collecting and curating data, designing the experiments and building the LLM training framework. The authors also participated in implementing and testing the experimental set-up, refining the RL process and performing the analysis of results. The scientific findings were discussed and approved by all contributors. This article was written by a subgroup of authors designated by the collaboration and it underwent an internal collective review process. All authors reviewed and approved the final version of the manuscript. Core contributors: D.G., D.Y., H.Z., J.S., R.Z., R.X., Q.Z., S.M., P.W., X.B., X.Z., X.Y., Y.W., Z.F.W., Z.G., Z.S., Z.L., Z.G. These authors, designated as core contributors, contributed equally to this work, and are presented in alphabetical order. The remaining authors have also contributed meaningfully to the study, and their names are likewise presented in alphabetical order.

**Competing interests** The authors declare no competing interests and will not file patents related to the content of this manuscript.

**Additional information**
**Correspondence and requests for materials** should be addressed to Wenfeng Liang.

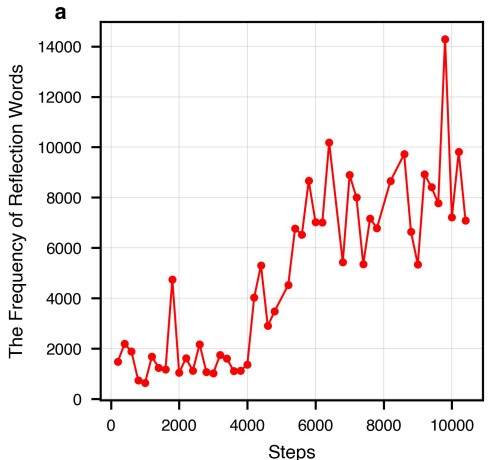

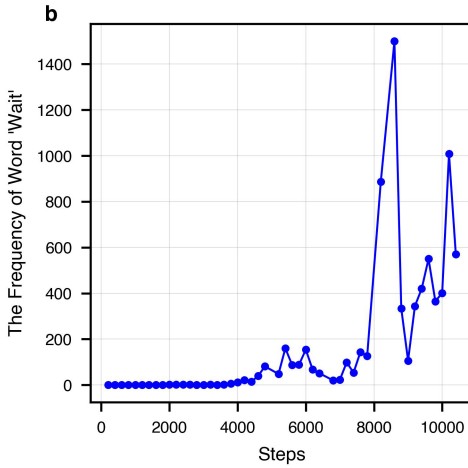

**Extended Data Fig. 1 | Evolution of reasoning-related linguistic features in model outputs across training steps. a**, Frequency of representative reflective terms in model-generated outputs throughout the training process. Reflective terms—including 'wait', 'mistake', 'however', 'but', 'retry', 'error', 'verify', 'wrong', 'evaluate' and 'check'—were identified and curated by a panel of three human experts. Each expert independently proposed a set of words indicative of reflective reasoning, which were subsequently consolidated through consensus into a final vocabulary list. **b**, Frequency of the term 'wait' in model outputs over the course of training. This term was virtually absent during the initial training stages, appeared sporadically between steps 4,000 and 7,000 and exhibited a marked increase in frequency after step 8,000. These trends suggest the emergence of temporal reasoning or self-monitoring behaviour as training progresses.

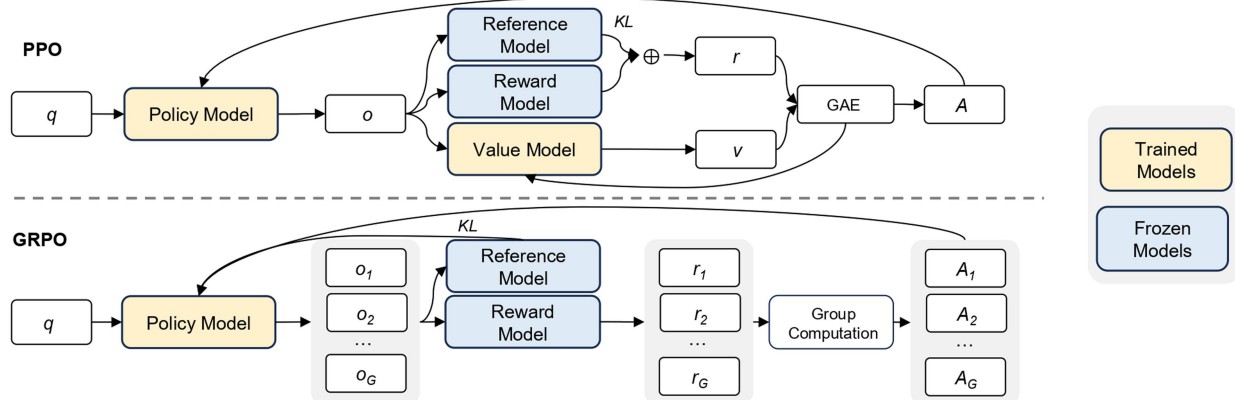

**Extended Data Fig. 2 | Illustration of the proposed GRPO for RL-based training.** In the proposed framework, a LLM is used as a policy model to generate responses $\{o_1, o_2,..., o_G\}$ conditioned on a given query $q$. Each response within the group is evaluated by a reward model—either learned (model-based) or manually specified (rule-based)—to assign a scalar reward signal. Subsequently, GRPO computes the relative advantages of each group member based on their assigned rewards. Rather than relying on an explicit value function, as in PPO, GRPO directly estimates advantages from the intra-group reward distribution. The policy parameters are then updated to maximize the expected reward while simultaneously minimizing divergence from a reference policy, typically quantified through the KL divergence. By eliminating the need for a separate value network, GRPO offers a simplified yet effective alternative to traditional actor-critic methods such as PPO.

**Extended Data Table 1 | An illustrative example from the AIME dataset**

### PROMPT

Let $b \geq 2$ be an integer. Call a positive integer $n$ *b-eautiful* if it has exactly two digits when expressed in base $b$, and these two digits sum to $\sqrt{n}$. For example, $81$ is $13$-eautiful because $81 = \underline{6}\ \underline{3}_{13}$ and $6 + 3 = \sqrt{81}$. Find the least integer $b \geq 2$ for which there are more than ten $b$-eautiful integers.

Please reason step by step, and put your final answer within \boxed{}.

### Evaluation

Parse the final answer within \boxed{} and use a rule-based grader to determine if it equals the ground truth. Round numerical values as needed, and use 'SymPy' to parse expressions.

The presented question is sourced from the 2024 AIME. The model is tasked with solving the problem and formatting its answer in a required format (for example, ANSWER). For evaluation, a rule-based grading system is used to determine correctness. The output of the model is considered correct if and only if it exactly matches the ground-truth solution.